# Rationale and design for studying organisation of care for intra-arterial thrombectomy in the Netherlands: simulation modelling study

Maarten M H Lahr  ,[1] Willemijn J Maas,[1,2] Durk-Jouke van der Zee,[3] Maarten Uyttenboogaart,[2,4] Erik Buskens[1,3]

For numbered affiliations see end of article.

**Correspondence to**
Dr Maarten M H Lahr;
m.m.h.lahr@umcg.nl

## ABSTRACT

**Introduction** The introduction of intra-arterial thrombectomy (IAT) challenges acute stroke care organisations to provide fast access to acute stroke therapies. Parameters of pathway performance include distances to primary and comprehensive stroke centres (CSCs), time to treatment and availability of ambulance services. Further expansion of IAT centres may increase treatment rates yet could affect efficient use of resources and quality of care due to lower treatment volume. The aim was to study the organisation of care and patient logistics of IAT for patients with ischaemic stroke in the Netherlands.

**Methods and analyses** Using a simulation modelling approach, we will quantify performance of 16 primary and CSCs offering IAT in the Netherlands. Patient data concerning both prehospital and intrahospital pathway logistics will be collected and used as input for model validation. A previously validated simulation model for intravenous thrombolysis (IVT) patients will be expanded with data of the MR CLEAN (Multicenter Randomized Clinical Trial of Endovascular Treatment for Acute Ischemic Stroke in the Netherlands) Registry and trials performed in the Collaboration for New Treatments in Acute Stroke consortium to represent patient logistics, time delays and outcomes in IAT patients. Simulation experiments aim to assess effectiveness and efficiency of alternative network topologies, that is, IAT with or without IVT at the nearest primary stroke centre (PSC) versus centralised care at a CSC. Primary outcomes are IAT treatment rates and clinical outcome according to the modified Rankin Scale. Secondary outcomes include onset-to-treatment time and resource use. Mann-Whitney U and Fisher's exact tests will be used to estimate differences for continuous and categorical variables. Model and parameter uncertainty will be tested using sensitivity analyses.

**Ethics and dissemination** This will be the first study to examine the organisation of acute stroke care for IAT delivery on a national scale using discrete event simulation. There are no ethics or safety concerns regarding the dissemination of information, which includes publication in peer-reviewed journals and (inter)national conference presentations.

**Trial registration number** ISRCTN99503308, ISRCTN76741621, ISRCTN19922220, ISRCTN80619088, NCT03608423; Pre-results.

## Strengths and limitations of this study

► The proposed simulation modelling study collects patient-level data from all intervention centres in the Netherlands that provide intra-arterial thrombectomy for patients with acute ischaemic stroke.
► Information from prehospital stroke services, including emergency medical services, is included in the model.
► Information on costs associated with pathway set-up and innovations foreseen in acute stroke treatment are included.
► Model results are estimations which have to be tested in clinical practice.
► Input parameters for model building contain estimations of time delays and diagnostic procedures that may have changed over time.

## INTRODUCTION

For patients with acute ischaemic stroke, reperfusion therapies comprise intravenous thrombolysis (IVT) administered up to 4.5 hours after onset and intra-arterial thrombectomy (IAT) administered up to 6 hours and in selected patients even up to 24 hours.[1–7] Functional recovery following treatment is strongly determined by the time interval between symptom onset and reperfusion. For every 9 min delay along the care pathway, 1 in every 100 patients treated with IAT has a worse disability outcome (higher modified Rankin Scale (mRS) score by one or more levels).[8] As such, timely arrival at the hospital and minimising the overall onset-to-treatment time are of critical importance. Following symptom onset, patients are typically transported to the nearest hospital capable of administering IVT, which can be either a PSC or a CSC. In parallel with administering the bolus for IVT, additional neuroimaging is performed in order to detect large-vessel occlusion (LVO). In case of an

LVO, patients may become eligible for IAT after which transfer to a CSC is arranged, if patients are initially admitted to a PSC.

The transfer of patients to a CSC following IVT diagnostic work-up and/or treatment at a PSC is called the drip-and-ship (DS) approach. In addition to the DS approach, patients may be transported directly to the CSC bypassing the PSC, which is called the mothership (MS) approach. Currently there are no formal policies supporting the latter, as it necessitates accurate prehospital triage objectifying LVO with sufficient accuracy. Timely patient arrival and treatment with reperfusion therapies depends on a number of factors such as the geographical distribution and distances between PSCs and CSCs in relation to the location of the patient. Arrangements made by local emergency medical services concerning referral patterns of suspected stroke victims and the quality and set-up of road networks also play a role, as does potential traffic congestion.

In the Netherlands, 16 medical centres that participated in the MR CLEAN trial are currently reimbursed for IAT. However, these hospitals are unevenly distributed, leaving certain regions potentially underserved, while in some areas there is a surplus of IAT centres. Insight into factors determining the potential optimal and most efficient way to organise acute stroke care is currently lacking but urgently needed to support clinicians and decision-makers. A key question is what would be the optimal distribution of CSCs within a region, taking into account the effects of additional travel distances, available clinical expertise, treatment volumes and capacity of hospitals? Are there regions underserved or overserved with CSCs and what would be the potential consequences of adding/removing CSCs within specific regions?

In this study, we aimed to quantify pathway logistics, that is, stroke onset and time delays associated with prehospital and intrahospital care services among patients treated with IAT with or without IVT. From this baseline assessment, a simulation model will be built that describes all activities and treatments performed along the acute stroke pathway. A generic model will be developed based on the set-up of IAT delivery in the region of Northern Netherlands and extended using results of the MR CLEAN registry[9] and data of five randomised clinical trials (RCTs) performed in the Collaboration for New Treatments in Acute Stroke (CONTRAST) consortium (www.contrast-consortium.nl). The MR CLEAN registry is an ongoing, prospective, observational study in all centres that perform IAT in the Netherlands. Within CONTRAST, five large clinical trials will be performed to test novel treatment strategies for stroke. This will be complemented by a preclinical programme, development of a large biobank, and identification of challenges and solutions for organisational models. Retrospective data from the MR CLEAN registry, combined with data from emergency medical services, will serve as input for the development of a baseline simulation model for regional IAT delivery. Next, prospective data capturing

using results from ongoing trials performed within CONTRAST will be used to further develop the model. Simulation experiments will be performed to assess effectiveness and efficiency of alternative network topologies, that is, primary and comprehensive stroke services (DS approach) versus a service delivery based on comprehensive and centralised care (MS approach). Model parameters will be extended by assigning unit costs to activities allowing for economic analyses. The latter will involve collection of detailed information on resource use allowing for microcosting studies and subsequent use in cost-benefit and/or cost-effectiveness analyses.

### Aims and hypotheses
*Primary aim:* to develop a simulation model for IAT in the Netherlands and to extend the model to reflect regional differences.

*Secondary aims:* to estimate the effectiveness of alternative network topologies on IVT and/or IAT treatment rates, geographical access to CSCs, time to treatment, treatment volumes and patient disability as measured by the mRS. Also, potential areas for pathway improvements in regional stroke care systems will be identified.

*Tertiary aims:* to extend the model with costs estimations of procedures and activities performed along the acute stroke pathway, allowing for elaborate economic evaluation. Also, the assessment and potential implementation of new technologies such as the mobile stroke unit is foreseen through modelling.

### METHODS
### Study design
This study uses discrete event simulation modelling in which care pathways can be represented 'in silico' by quantifying pathway performance.[10–12] In case of acute stroke treatment, information on time delays sustained by patients and diagnostic steps performed up to treatment with IVT and IAT are collected. In addition, patient demographics (age), exclusion and inclusion criteria, and follow-up data up to 3 months poststroke will be included. Previous research has demonstrated that simulation models for representing the acute stroke pathway can be accurately developed in different settings.[13–17] Typically, simulation modelling involves a number of consecutive steps, including model building, validation and experimentation. Model building starts with the conceptual representation of care pathways in building block, such as incidents, time delays, queues, resources and ultimately outcome. Next, the model needs to be populated through collection of empirical data reflecting actual pathway performance, for which a combination of real-world datasets is used. For hospital items, we will rely on case report forms (CRFs) collected in the RCTs in the CONTRAST consortium involving 16 hospitals in the Netherlands that provide IAT. In addition, collaboration with regional ambulance services will be set up in order to collect prehospital items, such as time of 911 call,

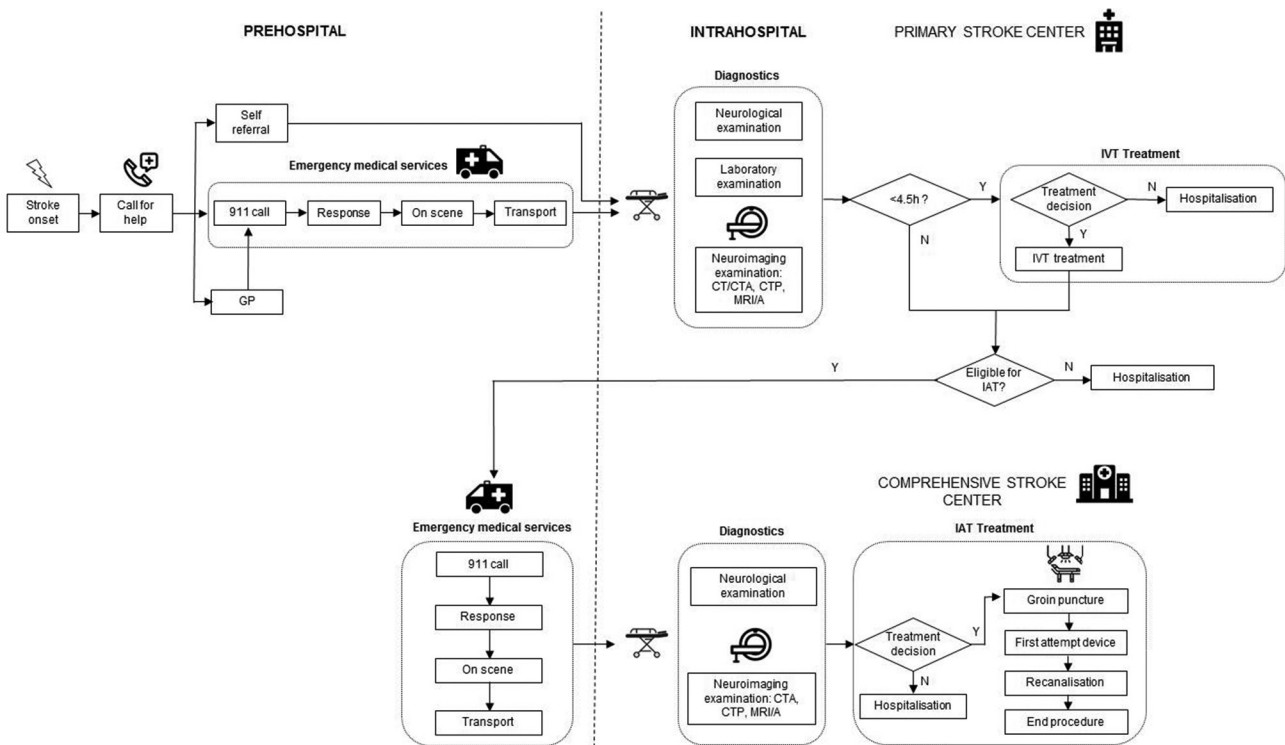

**Figure 1** Description of the drip-and-ship model. CTA, CT angiography; CTP, CT perfusion; GP, general practitioner; IAT, intra-arterial thrombectomy; IVT, intravenous thrombolysis; MRA, magnetic resonance angiography; N, no; Y, yes.

time spent on scene, transportation time to the hospital and general referral patterns. Real-world data collected in each step of the pathway will be analysed to distinguish statistical distributions using the statistical software package ExpertFit.[18] Based on the statistical distributions identified, hypothetical patients passing through the model will be assigned a certain time delay, and diagnostic accuracy for each step along the pathway. Model outcomes include the proportion of patients treated with IAT, time to treatment and functional outcome after 90 days. These results will then be validated comparing them to observed time delays and outcomes observed in clinical practice. Following model validation, experiments will be performed in which scenarios of alternative set-ups of the organisational models will be evaluated. Descriptions of the acute stroke pathway in both the DS and MS organisational models in Northern Netherlands are presented in figures 1 and 2.

## Study population

For this study, we will use data from patients with acute stroke who underwent IAT with or without IVT. A baseline model will be developed using a subset of patient information collected in the ongoing cohort study MR CLEAN Registry. Following the initial model patients treated in other regions in the Netherlands will be included. In later stages of model development, alternative treatment modalities may become apparent, such as treating patients with IAT up to 24 hours or including patients who underwent prehospital treatment in a mobile stroke unit. The acute stroke phase is defined as the time window in

which acute reperfusion therapies are being used. This can be the case up to a maximum of 24 hours following symptom onset. In these cases, the parameters underlying the model will be changed accordingly. Modelling the hypothetical introduction of a mobile stroke unit will involve changing prehospital stroke management. In such a scenario, an expedited and optimised process of initial diagnostic assessment and triage is assumed.

## Public and patient involvement

Patients and the public were involved in the conception of the topics to be addressed in the CONTRAST consortium. Study results will be disseminated through newsletters, poster presentations, and publications in newspapers, lay journals and publication in peer-reviewed journals.

## Data to be collected

An overview of all data items collected for the simulation model is presented in a table as supplementary material (online supplementary table S1, Overview of data collection). Based on these data, time intervals reflecting pathway efficiencies will be calculated, such as the time from symptom onset to CT/CT angiography (CTA) and door-to-CT/CTA times for patients arriving at a PSC and after secondary transfer to the CSC, if necessary.

### Hospital data

Within CONTRAST, five RCTs are performed that serve as input for the proposed simulation modelling studies: MR ASAP (NL60258.078.17), MR CLEAN LATE (NL58246.078.17), MR CLEAN MED (ISCRTN76741621),

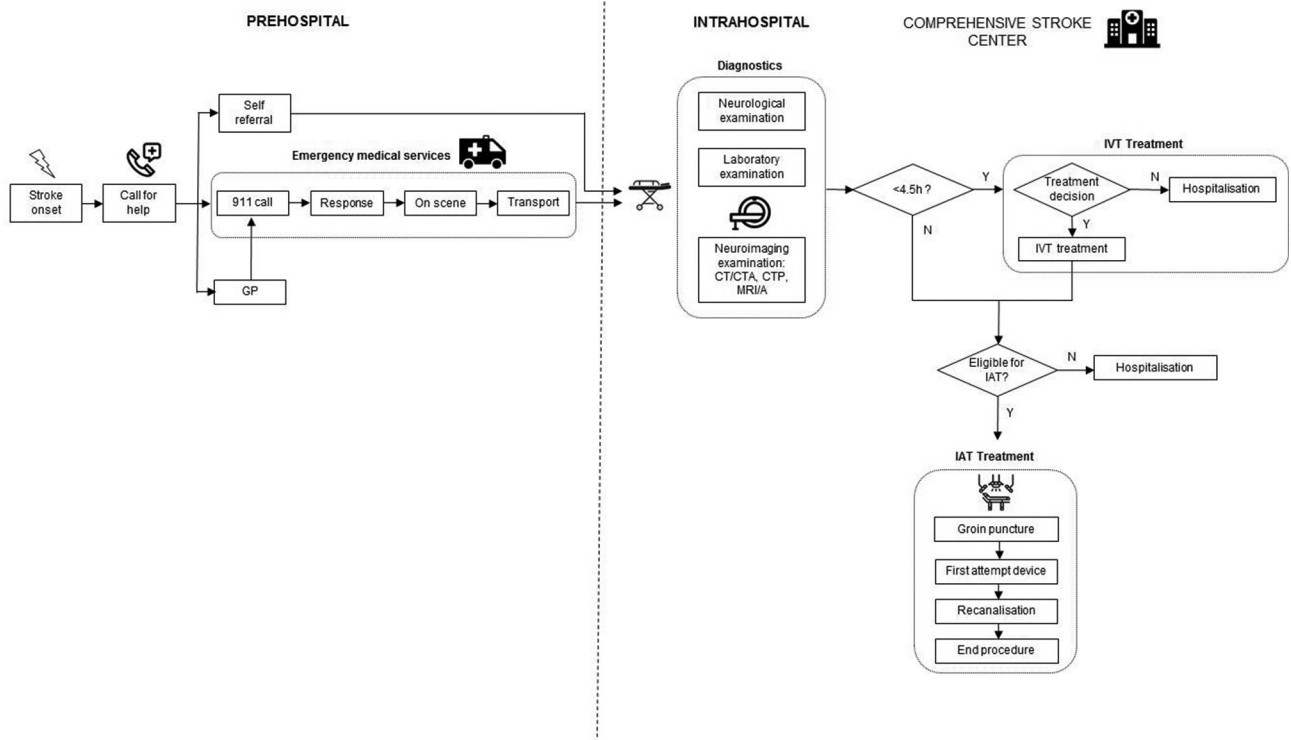

**Figure 2** Description of the mothership model. CTA, CT angiography; CTP, CT perfusion; GP, general practitioner; IAT, intra-arterial thrombectomy; IVT, intravenous thrombolysis; MRA, magnetic resonance angiography; N, no; Y, yes.

MR CLEAN NO IV (ISRCTN80619088). Each intervention centre uses a standardised, customised and web-based electronic CRF to document all steps in the care process. All data will be entered locally in a web-based database (OpenClinica). A copy of the CRF documents used in the clinical trials can be found online (www.contrast-consortium.nl). All studies aimed to collect a large set of items containing, among others, a description of the workflow, including time of symptom onset or last seen well, time of arrival of the first hospital and whether the patient was transferred from another hospital. Within the hospital, time to CT scan and advanced imaging (CTA/magnetic resonance angiography/digital subtraction angiography/perfusion imaging) are collected according to local procedures. Also, start of IVT (bolus infusion), patient arrival in the angiosuite, groin puncture, device attempts, recanalisation and sheath withdrawal/end of procedure are collected. In addition, anaesthetic team presence and management procedures are recorded.

### Prehospital data

For the collection of prehospital data collaboration with regional ambulance services is being set up. Items will include time delays, diagnostics and geographical information on referral patterns. Time delays include the time of 911 call, arrival at the location of the patient, departure at the location of the patient and arrival at the hospital. Diagnostic items include mode of referral (general practitioner, 911 or other), initial working diagnosis set by 911 dispatch and ambulance personnel, and the level of urgent transportation (ie, blue lights or other).

Geographical information includes the postal codes of the location of the patient, departure location of the ambulance vehicle, and location of the PSC and/or CSC.

### Baseline data

Prior to the CONTRAST study, all hospitals contributed to the MR CLEAN Registry, an online and ongoing database to monitor implementation and safety of IAT in the Netherlands.[9] For CONTRAST, the variables are similar to those collected in the MR CLEAN Registry. In addition, prehospital information available at regional ambulance services will be collected.

The baseline discrete event simulation model will be developed using data from 296 patients who underwent IAT at the University Medical Centre Groningen between July 2014 and November 2017. For these patients, intra-hospital time delays and diagnostic steps were already collected as part of the ongoing MR CLEAN Registry and were subsequently used for model development and validation. In addition, prehospital items as described previously were collected in collaboration with regional ambulance services.

### Outcome measures

Outcomes will be estimated by the simulation model using parameter distributions as observed in clinical practice using real-world data.

*Primary outcomes* include IAT treatment rates and clinical outcome according to the mRS.

*Secondary outcomes* include process times, such as symptom onset to hospital arrival, door to needle and

groin puncture, arrival angiosuite, recanalisation, onset-to-treatment time of IVT and/or IAT, and interhospital transportation times for those patients who were transferred to a CSC after initial diagnosis and/or treatment with IVT at a PSC. In addition, the time interval of referral from the PSC to CSC is included.

*Tertiary outcomes* include treatment volume, direct costs, indirect costs and societal costs. This will include fixed and variable costs of providing services related to IAT delivery.

## Ethics and dissemination

Both the MR CLEAN and CONTRAST databases include patients for which informed consent had been obtained, which was extended for use in our simulation modelling study. All data used for model building have been completely anonymised, performed by local researchers under auspices of the principal investigators for the MR CLEAN Registry and CONTRAST studies. Accordingly, no additional approval from our local ethics committee is required, as this represents a simulation study using an anonymised dataset.

In order to link data collected of patients treated with IAT after study enrolment in the hospital to corresponding data collected by ambulance services, identifiable information from patients will be used under a strict protocol compliant with the Dutch Personal Data Protection Act. This protocol for using prehospital ambulance data meets the criteria for a waiver of consent from the University Medical Centre Groningen Institutional Review Board, for which formal approval was already obtained. Following successful linking of prehospital and inhospital data, note, however, that also these data were anonymised. Prior to the exchange of information, a data transfer and processing agreement was agreed on and signed by each provider of regional ambulance services.

Study results will be published in peer-reviewed journals and (inter)national conference presentations. Centres that have participated in the study will have access to their own data and results.

## Study sites

Sixteen hospitals providing IAT will participate. In addition, regional ambulance services in the catchment area of intervention centres will be asked to participate. The overview of intervention centres, their catchment area and regional ambulance services active in the region are provided as supplementary material (online supplementary figure S1, Overview of IAT centres and ambulance services).

## Statistical analyses

Descriptive statistics will be used to characterise the population and regional differences. Mann-Whitney U and Fisher's exact tests are used to estimate the differences for continuous and categorical variables. Model and parameter uncertainty will be tested using sensitivity analyses.

## Approach to missing data

Multiple imputation techniques will be used to handle missing data. The impact of missing data will be analysed by performing sensitivity analyses. Once missing data or errors are identified, all corrections made will be documented. All data will be documented and displayed in an appropriate flow diagram.

## Model validation

Internal validation will be performed by comparing model performance with observed performance of the stroke system to ascertain whether the model represented the real system accurately. External validation of the baseline model will be performed by using the pathway performance of other PSCs and CSCs participating in the CONTRAST consortium. As such, the model developed for one region will be extended to other parts of the Netherlands by changing the distributions underlying the simulation model. This means that the model is repopulated by real-world data originating from different regions, including both intrahospital and prehospital data. As we will adopt one uniform method for data collected in both the prehospital and intrahospital pathways, similar procedures for model building, validation and experimentation can be followed.

## Study organisation

The CONTRAST consortium is a nationwide collaboration of clinical and translational scientists from all academic and large clinical centres who want to act together to improve the treatment of acute stroke in the Netherlands. Five large acute stroke trials to test novel treatment strategies will be performed, including prehospital augmentation of collateral blood flow and blood pressure management (MR ASAP), antithrombotics to prevent microvascular occlusion after IAT (MR CLEAN MED), immediate IAT without preceding thrombolysis (MR CLEAN NO IV), IAT in the time window of 6–24 hours (MR CLEAN LATE) and minimally invasive endoscopy-guided surgery for intracerebral haemorrhage (Dutch ICH trial). Discrete event simulation modelling is applied with data from these trials to optimise acute stroke care delivery.

## DISCUSSION

This simulation modelling study will evaluate the set-up of acute stroke services for patients undergoing IAT in a specific geographical setting, that is, the Netherlands. However, the model foreseen may also serve as a planning and evaluation tool for other regions and countries. Already from an early phase onwards, the potential consequences of the clinical trials performed in CONTRAST on the organisation of acute stroke care will be assessed. Benefits of innovations foreseen in prehospital as well as hospital-based diagnosis and treatment will be estimated on a regional and national scale. Likewise, the optimal organisation ensuing from these reorganising care

processes may be identified and taken into account, given the available resources and capacity. Another important aspect that will be studied is how redistribution of PSCs and CSCs within certain regions will affect time to treatment for both IVT and IAT.[19] Experiments from simulation models will help to elucidate factors determining optimal stroke pathway set-up and to identify potential targets for further optimisation of pathway performance. Simulation modelling also allows for making exact estimations on effects on patients health and costs when adding or removing IAT centres in specific regions, thereby supporting clinicians and policy makers in decision-making.[15]

Our simulation modelling approach has several strengths. First, the prospective design and uniform methods for data collection both in the prehospital and intrahospital phases will ensure a high quality of data capture. The use of real-world data will support the external validation of results obtained and is a unique feature of the proposed modelling study. Also, in terms of efficiency, simulation modelling is an attractive option as it uses already collected information in clinical trials and by ambulance services. Second, the model will have a comprehensive scope, contrary to current improvement approaches characterised by monodimensional interventions.[20–22] Implementation problems such as how many CSCs would a region need or where they should ideally be located require a multifaceted solution, and could therefore benefit from a modelling approach. Data used for model input will be collected in various regions of the Netherlands reflecting differences in access to PSCs and CSCs, road networks and potential traffic congestion. Third, the model builds on a previously developed simulation model for IVT treatment in Northern Netherlands.[10] As such, we have demonstrated that model building and validation are feasible. Finally, the model can be extended to include unit costs for elaborate economic analyses. The comprehensive character of the modelling approach also allows considering potential interaction effects between variables and has a quadruple dimensional approach in terms of measuring outcomes, as it can capture effects on clinical outcomes, time to treatment, referral patterns and costs of processes associated with IVT and IAT delivery.

Serving as a precursor for clinical validation, results obtained from modelling studies should always be interpreted with care as these results do not represent actual patients. However, from an economics and business approach, each treatment can be represented as the sum of time delays and diagnostics performed along the care pathway, and as such can be quantified and in turn represented in silico by computer models. One of the main questions is whether there is benefit in bypassing the nearest PSC to transport the patient directly to a CSC. At the moment, there is no consensus which prehospital scale to use in order to quantify stroke severity, and no agreed on triage instrument exists to guide transfer of patients to specific hospitals. Typically, in case of a positive trial result in CONTRAST, this will have a direct impact on the organisational model for IAT delivery, either by suggesting to bypass PSCs (MR CLEAN NO IV) by extending the time window for reperfusion therapies (MR CLEAN LATE) or to change prehospital stroke management (MR ASAP). As a consequence, alternative set-up of services is foreseen, in which accurate estimations on effectiveness, time to treatment and cost-effectiveness are required to support decision-making.

In summary, the proposed modelling study will investigate current organisational models for IAT delivery in the Netherlands. Based on input from real-world clinical studies performed in the CONTRAST consortium insight into current treatment delays and workflow performance of IAT centres will be obtained. Next, simulation modelling studies will support design of optimal distribution of IAT centres, in order to improve patient outcomes and increase the proportion of patients eligible for IAT.

**Author affiliations**
[1]Health Technology Assessment, Department of Epidemiology, University of Groningen, University Medical Centre Groningen, Groningen, The Netherlands
[2]Department of Neurology, University of Groningen, University Medical Centre Groningen, Groningen, The Netherlands
[3]Department of Operations, Faculty of Economics & Business, University of Groningen, Groningen, The Netherlands
[4]Department of Radiology, University of Groningen, University Medical Centre Groningen, Groningen, The Netherlands

**Contributors** MMHL, WJM, D-JvdZ, MU and EB designed the study with MU and EB as principal investigators. MMHL, MU and EB applied for, received and organised the study funding. MMHL drafted the manuscript. WJM, D-JvdZ, MU and EB critically revised the manuscript for intellectual content and approved the final version of the manuscript for publication.

**Funding** The Collaboration for New Treatments in Acute Stroke consortium is supported by Netherlands Cardiovascular Research Initiative, an initiative of the Dutch Heart Foundation, by the Brain Foundation Netherlands and powered by Health~Holland, Top Sector Life Sciences, and receives unrestricted funding from Medtronic.

**Competing interests** None declared.

**Patient consent for publication** Not required.

**Provenance and peer review** Not commissioned; externally peer reviewed.

**ORCID iD**
Maarten M H Lahr http://orcid.org/0000-0001-7265-2612

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
