## [Reviewer comments · BMJ Open]

ARTICLE DETAILS

TITLE (PROVISIONAL)	Rationale and design for studying organisation of care for intra-arterial thrombectomy in the Netherlands: a simulation modelling study
AUTHORS	Lahr, Maarten; Maas, Willemijn; van der Zee, Durk-Jouke; Uyttenboogaart, Maarten; Buskens, Erik

VERSION 1 - REVIEW

REVIEWER	Michael Allen University of Exeter Medical School
REVIEW RETURNED	01-Aug-2019

GENERAL COMMENTS	This is probably the most comprehensive collection of data undertaken in stroke pathway modelling to date. Good luck - I will look forward to seeing the results!
---

REVIEWER	Prof. Alastair Buchan Radcliffe Department of Medicine University of Oxford United Kingdom
REVIEW RETURNED	25-Sep-2019

GENERAL COMMENTS	Comments for Authors Manuscript Nr.: bmjopen-2019-032754 Author for Correspondence: Dr. Maarten Lahr Title: Rationale and design for studying organization of care for intra-arterial thrombectomy in the Netherlands: a simulation modelling study The proposed study protocol aims to study the organizational models and patient logistics of intra-arterial thrombectomy at sixteen Dutch stroke centers for ischemic stroke patients with the main goal of improving patient outcomes and increasing the proportion of patients eligible for intra-arterial thrombectomy (IAT). The three study aims are as follows: (1) To develop a simulation model for IAT and extend the model to reflect regional differences; (2) To estimate the difference in treatment outcome based on topologies of intra-venous thrombectomy (IVT) versus IAT and find areas where treatment access shall be improved; (3) To evaluate
---

the economic cost of the current model and potential interventions such as the mobile stroke unit.

Publication of such study protocols aiming to improve patients outcomes and logistics in therapeutic delivery is of great importance. The study is very timely with its main question of reorganizing stroke services to improve patient care. Further, the illustration of the drip-and-ship and mothership model are very well done (Figures 1 and 2). Minor corrections are needed for its reaching a suitable level for publication in BMJ Open.

Minor issues:

- Abstract: Change patient "disability" to "clinical outcome".
 - Introduction: Further explain the type of economic analysis (Microcosting vs. Macrocosting vs. Cost-effectiveness analysis).
 - Introduction: Define the MR CLEAN and CONTRAST studies.
- Further elucidate the source of the clinical data to be used: Is it retrospective data from MR CLEAN/CONTRAST studies or would it prospective data capturing real time and resources?
- Aims and Hypothesis: Mrs has autocorrected and should be "mRS".
 - Aims and Hypothesis, Tertiary aims: Please comment on how the information from the mobile stroke unit would be used.
 - Study design: Inclusion of information on patient demographics (age), exclusion and inclusion criteria, as well as follow-up data of at least three months post stroke would be of great importance.
 - Study design: Write "in silico" in italics.
 - Study design: Change "truthfully" for "accurately".
 - Public and patient involvement: Change "lay-men" journals to "lay" journals
 - Data to be collected: If possible, please include the onset-to-CT/CTA and door-to-CT/CTA time
 - Outcome measures, Secondary outcomes: Consider the inclusion of the time of referral from the primary stroke center (PSC) to comprehensive stroke center (CSC).
 - Outcome measures, Tertiary outcomes: Further elucidate which aspect of the societal costs will be studied.
 - Approach to missing data: State that all data will be documented and displayed in an appropriate flow diagram.
 - Discussion: Change "data capturing" for "data capture".
 - Discussion: Please rephrase the following sentence: "Proposed solutions following such an approach are per definition finite, and cannot not solve implementation problems such as how many CSCs would a region need or where they should ideally be located."
 - Discussion: Consider discussing the research findings of Allen et al. on how to maximize thrombectomy access for ischemic stroke patients in England. (Allen M, et al. Maximising access to thrombectomy services for stroke in England: A modelling study. Stroke 2019; 4(1) 39-49
- A modelling study. Stroke 2019; 4(1) 39-49
- General: Please clarify the time of the acute stroke phase: 24/48/72h?
 - Conclusion: Please consider renaming the conclusion section to impact summary or similar. Based on the notes from the editors, no results or conclusion shall be included in the manuscript.
 - Funding: How can the authors assure that there is no conflict of interest because of the study's financial support by Medtronic?

VERSION 1 – AUTHOR RESPONSE

Michael Allen (Reviewer 1): This is probably the most comprehensive collection of data undertaken in stroke pathway modelling to date. Good luck - I will look forward to seeing the results!

Response of the authors: we thank the reviewer for his kind comment.

Prof. Alastair Buchan (Reviewer 2): The proposed study protocol aims to study the organizational models and patient logistics of intra-arterial thrombectomy at sixteen Dutch stroke centers for ischemic stroke patients with the main goal of improving patient outcomes and increasing the proportion of patients eligible for intra-arterial thrombectomy (IAT). The three study aims are as follows: (1) To develop a simulation model for IAT and extend the model to reflect regional differences; (2) To estimate the difference in treatment outcome based on topologies of intra-venous thrombectomy (IVT) versus IAT and find areas where treatment access shall be improved; (3) To evaluate the economic cost of the current model and potential interventions such as the mobile stroke unit.

Publication of such study protocols aiming to improve patients outcomes and logistics in therapeutic delivery is of great importance. The study is very timely with its main question of reorganizing stroke services to improve patient care. Further, the illustration of the drip-and-ship and mothership model are very well done (Figures 1 and 2). Minor corrections are needed for its reaching a suitable level for publication in BMJ Open.

Comment 1: Abstract: Change patient “disability” to “clinical outcome”.

Response of the authors:

We agree with the comment of the reviewer. Therefore we changed the sentence accordingly. It now reads: “..clinical outcome”. [Abstract, page 2, line 18; Outcome measure, page 7, line 35]

Comment 2: Introduction: Further explain the type of economic analysis (Microcosting vs. Macrocosting vs. Cost-effectiveness analysis).

Response of the authors:

We thank the reviewer for this valuable comment. Therefore we added an extra sentence explaining the type of economic analysis performed. It now reads: “The latter will involve collection of detailed information on resource use allowing for micro-costing studies and subsequent use in cost-benefit and/or cost-effectiveness analyses”. [Introduction, page 5, line 6-8]

Comment 3: Introduction: Define the MR CLEAN and CONTRAST studies.

Response of the authors:

We thank the reviewer for raising this point. Therefore we added two sentences to define both studies. It now reads: "The MR CLEAN registry is an ongoing, prospective, observational study in all centres that perform IAT in the Netherlands. Within CONTRAST 5 large clinical trials will be performed to test novel treatment strategies for stroke. This will be complemented by a pre-clinical program, development of a large biobank and the identification of challenges and solutions for organisational models". [Introduction, page 4, line 41-46]

Comment 4: Further elucidate the source of the clinical the data to be used: Is it retrospective data from MR CLEAN/CONTRAST studies or would it prospective data capturing real time and resources?

Response of the authors:

We agree with the reviewer and are grateful for pointing this out. Therefore we added a sentence to clarify the issue raised. It now reads: "Retrospective data from the MR CLEAN registry combined with data from emergency medical services will serve as input for the development of a baseline simulation model for regional IAT delivery. Next, prospective data capturing using results from ongoing trials performed within CONTRAST will be used to further develop the model". [Introduction, page 4, line 46-48; page 5, line 1-2]

Comment 5: Aims and Hypothesis: Mrs has autocorrected and should be "mRS".

Response of the authors:

We thank the reviewer for pointing out this mistake, and changed it accordingly. It now reads: "mRS". [Aims and hypotheses, page 5, line 15; Outcome measures, page 7, line 36]

Comment 6: Aims and Hypothesis, Tertiary aims: Please comment on how the information from the mobile stroke unit would be used.

Response of the authors:

We thank the reviewer for this valuable comment. Therefore we added a sentence to clarify the issue raised. It now reads: "Modelling the hypothetical introduction of a mobile stroke unit will involve changing prehospital stroke management. In such a scenario an expedited and optimised process of initial diagnostic assessment and triage is assumed". [Study population, page 6, line 17-19]

Comment 7: Study design: Inclusion of information on patient demographics (age), exclusion and inclusion criteria, as well as follow-up data of at least three months post stroke would be of great importance.

Response of the authors:

We agree with the comment of the reviewer. Therefore we added a sentence to include this information. It now reads: "In addition, patient demographics (age), exclusion and inclusion criteria and follow-up data up to three months post stroke will be included". [Study design, page 5, line 27-29]

Comment 8: Study design: Write "in silico" in italics.

Response of the authors:

We agree with the comment of the reviewer, and changed the text accordingly. It now reads: "...in silico". [Study design, page 5, line 25; Discussion, page 10, line 5]

Comment 9: Study design: Change "truthfully" for "accurately".

Response of the authors:

We agree with the comment of the reviewer, and changed the text accordingly. It now reads: "...accurately". [Study design, page 5, line 30]

Comment 10: Public and patient involvement: Change "lay-men" journals to "lay" journals

Response of the authors:

We agree with the comment of the reviewer, and changed the text accordingly. It now reads: "...lay journals". [Public and patient involvement, page 6, line 24]

Comment 11: Data to be collected: If possible, please include the onset-to-CT/CTA and door-to-CT/CTA time

Response of the authors:

We agree with the comment of the reviewer. Therefore we added a sentence to this section. It now reads: "Based on this data, time intervals reflecting pathway efficiencies will be calculated, such as the time from symptom onset to computed tomography (CT)/ computed tomography angiography (CTA) and door-to-CT/CTA times for patients arriving at a PSC and after secondary transfer to the CSC, if necessary". [Data to be collected, page 6, line 29-33]

Comment 12: Outcome measures, Secondary outcomes: Consider the inclusion of the time of referral from the primary stroke center (PSC) to comprehensive stroke center (CSC).

Response of the authors:

We thank the reviewer for raising this important point. Therefore we added a sentence to this section. It now reads: "In addition, the time interval of referral from the PSC to CSC is included". [Outcome measures, page 7, line 40-41]

Comment 13: Outcome measures, Tertiary outcomes: Further elucidate which aspect of the societal costs will be studied.

Response of the authors:

We thank the reviewer for raising this issue. Therefore we added a sentence to this section. It now reads: "This will include fixed and variable costs of providing services related to IAT delivery". [Outcome measures, page 7, line 43]

Comment 14: Approach to missing data: State that all data will be documented and displayed in an appropriate flow diagram.

Response of the authors:

We thank the reviewer for this comment. Therefore we added a sentence to this section. It now reads: "All data will be documented and displayed in an appropriate flow diagram". [Approach to missing data, page 8, line 25-26]

Comment 15: Discussion: Change "data capturing" for "data capture".

Response of the authors:

We thank the reviewer for this comment, and changed the text accordingly. It now reads: "...data capture". [Discussion, page 9, line 28]

Comment 16: Please rephrase the following sentence: "Proposed solutions following such an approach are per definition finite, and cannot not solve implementation problems such as how many CSCs would a region need or where they should ideally be located".

Response of the authors:

We thank the reviewer for his comment. Therefore we rephrased the sentence. It now reads: "Implementation problems such as how many CSCs would a region need or where they should ideally be located require a multi-faceted solution, and could therefore benefit from a modelling approach". [Discussion, page 9, line 33-35]

Comment 17: Discussion: Consider discussing the research findings of Allen et al. on how to maximize thrombectomy access for ischemic stroke patients in England. (Allen M, et al. Maximising access to thrombectomy services for stroke in England: A modelling study. Stroke 2019; 4(1) 39-49

Response of the authors:

We thank the reviewer for his suggestion. Therefore we decided to include the research findings of Allen et al. in the Discussion section. It now reads: "Another important aspect that will be studied is how re-distribution of PSCs and CSCs within certain regions will affect time to treatment for both IVT and IAT". [Discussion, page 9, line 19-21]

Comment 18: General: Please clarify the time of the acute stroke phase: 24/48/72h?

Response of the authors:

We thank the reviewer for his comment. Therefore we added a sentence to clarify the issue raised. It now reads: "The acute stroke phase is defined as the time window in which acute reperfusion therapies are being used. This can be the case up to up to a maximum of 24 hours following symptom onset". [Study population, page 6, line 13-16]

Comment 19: Conclusion: Please consider renaming the conclusion section to impact summary or similar. Based on the notes from the editors, no results or conclusion shall be included in the manuscript.

Response of the authors:

We thank the reviewer for his suggestion, and changed the section accordingly. It now reads: "In summary, .." [Discussion, page 10, line 18]

Comment 20: Funding: How can the authors assure that there is no conflict of interest because of the study's financial support by Medtronic?

Response of the authors:

We thank the reviewer for raising this important issue. Medtronic provides unrestricted study funding for the CONTRAST consortium, yet, the present study on organizational issues in acute stroke care is funded by the Netherlands Heart Foundation. [Funding, page 10, line 36-39]. For the clinical trials Medtronic is given up to two weeks to comment on a manuscript prior to submission. However, the scientific committee is responsible for the final version of the reports.

VERSION 2 – REVIEW

REVIEWER	Prof. Alastair Buchan University of Oxford, UK
REVIEW RETURNED	07-Nov-2019

GENERAL COMMENTS	Comments for Authors Manuscript Nr.: bmjopen-2019-032754, R1 Author for Correspondence: Dr. Maarten Lahr Title: Rationale and design for studying organization of care for intra-arterial thrombectomy in the Netherlands: a simulation modelling study The proposed study protocol aims to study the organizational models and patient logistics of intra-arterial thrombectomy at sixteen Dutch stroke centers for ischemic stroke patients with the main goal of improving patient outcomes and increasing the proportion of patients eligible for intra-arterial thrombectomy (IAT). The three study aims are as follows: (1) To develop a simulation model for IAT and extend the model to reflect regional differences; (2) To estimate the difference in treatment outcome based on topologies of intra-venous thrombectomy (IVT) versus IAT and find areas where treatment access shall be improved; (3) To evaluate the economic cost of the current model and potential interventions such as the mobile stroke unit. Publication of such study protocols aiming to improve patients outcomes and logistics in therapeutic delivery is of great importance. The study is very timely with its main question of reorganizing stroke services to improve patient care. Further, the illustration of the drip-and-ship and mothership model are very well done (Figures 1 and 2). The authors responded to and implemented all the reviewer's suggestions for improvement. We find this manuscript has reached a suitable level for publication in BMJ Open and we therefore suggest acceptance.
---